# Soluble Pecam-1 as a Biomarker in Periprosthetic Joint Infection

**DOI:** 10.3390/jcm10040612

**Published:** 2021-02-05

**Authors:** Michael Fuchs, Andrej Trampuz, Stephanie Kirschbaum, Tobias Winkler, F Andrea Sass

**Affiliations:** 1RKU University Department of Orthopaedics, University of Ulm, 89081 Ulm, Germany; michael.fuchs@rku.de; 2Center for Musculoskeletal Surgery, Charité—University Medicine Berlin, 10117 Berlin, Germany; andrej.trampuz@charite.de (A.T.); stephanie.kirschbaum@charite.de (S.K.); tobias.winkler@charite.de (T.W.); 3Julius Wolff Institute, Charité—University Medicine Berlin, 13353 Berlin, Germany; 4Berlin Institute of Health Center for Regenerative Therapies, Charité—University Medicine Berlin, 13353 Berlin, Germany

**Keywords:** biomarker, TKA, periprosthetic joint infection, PJI, revision arthroplasty, diagnostic tool

## Abstract

A reliable workup with regard to a single diagnostic marker indicating periprosthetic joint infection (PJI) with sufficient sensitivity and specificity is still missing. The immunologically reactive molecule Pecam-1 is shed from the T-cell surface upon activation via proinflammatory signaling, e.g., triggered by specific pathogens. We hypothesized that soluble Pecam-1 (sPecam-1) can hence function as a biomarker of PJI. Fifty-eight patients were prospectively enrolled and assigned to one of the respective treatment groups (native knees prior to surgery, aseptic, and septic total knee arthroplasty (TKA) revision surgeries). Via synovial sample acquisition and ELISA testing, a database on local sPecam-1 levels was established. We observed a significantly larger quantity of sPecam-1 in septic (*n* = 22) compared to aseptic TKA revision surgeries (*n* = 20, *p* ≤ 0.001). Furthermore, a significantly larger amount of sPecam-1 was found in septic and aseptic revisions compared to native joints (*n* = 16, *p* ≤ 0.001). Benchmarking it to the gold standard showed a high predictive power for the detection of PJI. Local sPecam-1 levels correlated to the infection status of the implant, and thus bear a strong potential to act as a biomarker of PJI. While a clear role of sPecam-1 in infection could be demonstrated, the underlying mechanism of the molecule’s natural function needs to be further unraveled.

## 1. Introduction

With increasing life expectancy and concurrent high demands regarding personal mobility, the numbers of total joint arthroplasties are rising [1,2,3]. In arthroplasty patients, periprosthetic joint infection (PJI) still constitutes a major challenge with potentially devastating complications. As one of the main reasons for implant failure, PJI occurs in 0.3–4% of all primary hip and knee arthroplasties, with even higher rates of up to 15% in revision surgeries [4,5].

To date, the diagnosis of PJI is still based on various examinations, taking into account individual serum CRP levels, synovial leucocyte counts, and potential microbial identifications of bacterial pathogens, as well as histopathological examinations [3,6,7]. While the existing PJI criteria systems lead to fair results, the diagnostic of persistent low-grade infections are insufficient, calling for the development of highly sensitive tools for early detection. Despite established evaluations of serum inflammatory markers, synovial fluid analysis can be seen as the diagnostic mainstay [8]. Within the current literature, many studies focus on new biomarkers for a reliable detection of PJI [2,9,10]. Nevertheless, there is no single diagnostic tool with sufficient sensitivity and specificity.

While infection treatment focuses mainly on diagnostics of bacterial specimen, the endogenous immunological competence of patients and its connection to the patients’ likeliness to develop periprosthetic infections has so far not been getting enough attention. Previous research highlighted the role of T-cells and the adaptive immune system in PJI [7,11,12]. Pecam-1 positive T-cells have been described as a valid marker for thymus activity and can hence be seen as an indicator for the individual’s immunological status [13]. On a cellular level, Pecam-1 has a broad range of immunoregulatory functions, such as T-cell activation control and survival, dampening of pro-inflammatory cytokine production, and prevention of macrophage phagocytosis. Particularly in T-cells, Pecam-1 plays a central role, since this immunologically reactive molecule is shed from the surface of naive T-cells upon activation (Figure 1). The molecule is homophilic, capable of homo-oligomerization, and potentially competitive to bound protein [14,15].

The serum of septic shock patients has previously been shown to contain significantly higher sPecam-1 levels than control patients, leading to the pursuing question on the role of cell-shed sPecam-1 in PJI due to an intrinsic T-cell-induced activation by specific pathogens [15]. It remains unclear if cell-shed sPecam-1 is locally produced in order to fight an infection or inflammation (which is certainly not to be equated), or if it is a by-product that may lead to unfavorable cytotoxicity.

We hypothesized that soluble Pecam-1 can not only be used to assess the level of T-cell activation and reactivity of the adaptive immune response, but can function locally as a biomarker for lingering infections. Thus, this study aims to prove the potential of sPecam-1 as a marker for PJI.

## 2. Material and Methods

### 2.1. Study Design

This prospective study investigated the potential of sPecam-1 as a marker for PJI of the knee. Samples were prospectively collected from 1 August 2016 to 31 July 2017 in a unit for musculoskeletal and periprosthetic infections in one academic center. The study protocol was approved by the local ethics committee (EA1/033/17).

### 2.2. Patients

Patients were prospectively identified in our outpatient department. Consecutive patients aged ≥18 years were screened for inclusion. Synovia samples were taken intraoperatively via joint aspiration after skin incision and subcutaneous preparation. Samples were immediately aliquoted, stored at −80 °C, and measured within 21 days after surgery. Patients were assigned to one of the three defined intervention groups depending on the surgical procedure: (A) native knees prior to total knee arthroplasty (TKA), (B) aseptic revision TKA, (C) septic revision surgery with either debridement, antibiotics, and implant retention (DAIR), 1-stage TKA revision surgery, or 2-stage revision surgery via interposition of a static spacer followed by staged TKA reimplantation. The group’s assignment was made according to the EBJIS-proposed criteria of PJI as visualized in Figure 2. Definition of group A, B, and C are hence as follows:(A) Native knees prior to TKA:

This cohort served as a control group defining the individual sPecam-1 levels under aseptic conditions in native joints. Patients with prior arthroscopic or open interventions adjacent to the knee joint were excluded from further evaluation. Additionally, patients were screened pre-operatively for clinical or laboratory signs of infection. Patients with an elevated serum C-reactive protein level > 7 mg/L and with signs of redness or swelling of the affected limb were excluded.

(B) Aseptic revision knee replacement surgery:

Aseptic revision surgery was performed in patients without suspected PJI. The differentiation of aseptic and septic revision surgery was determined according to the EBJIS criteria (Figure 2) [7,16].

(C) Septic revision knee replacement surgery:

Patients with clinical and laboratory diagnostics indicating PJI according to the EBJIS criteria were assigned to the septic revision study cohort. Furthermore, these entities were differentiated in acute and chronic infections according to Izakovicova et al [17]. Acute infections were defined as early postoperative infections < 4 weeks, or acute hematogenous infections with duration of the symptoms < 3 weeks. Acute PJI are caused by high-virulent pathogens, such as *S. aureus* or Gram-negative bacteria, and are accompanied by clinical features highly suspicious for infection [17]. In patients with acute PJI, debridement, antibiotics, and implant retention (DAIR) were performed as the treatment of choice. Chronic PJI presents ≥4 weeks after surgery or with ≥3 weeks of duration of symptoms. Typical clinical features of acute PJI such as acute pain and redness of the swollen joint are often absent in chronic infections. Instead, patients with these pathologies may suffer from pain and loosening of the prosthesis. Chronic PJI are caused by low-virulent pathogens such as coagulase-negative *Staphylococcus* and *Cutibacterium* species [17,18]. These patients were treated with septic 1- or 2-stage revision surgery after complete removal of the prosthesis.

### 2.3. Quantitative Analyses

Via large-scale sample acquisition of freshly collected synovial fluid from primary, aseptic, and septic revision knee surgeries, a database on local sPecam-1 quantities in correlation with the infection status of the patient was established. sPecam-1 quantities in synovia were measured via ELISA (human sPecam-1 ELISA, eBioscience).

### 2.4. Cytotoxicity

The cytosolic enzyme lactate dehydrogenase (LDH) is released into the cell culture medium upon damage to the plasma membrane and can hence function as a marker for cytotoxicity of a tested substance. LDH assays were performed to preclude potential cytotoxic effects of s-Pecam-1 in different physiological quantities, corresponding to the median calculated from native, septic, and aseptic patients as measured via ELISA (native as in low sPecam-1 = 26 ng/mL, aseptic as in medium sPecam-1 = 44 ng/mL, septic as in high sPecam-1 = 73 ng/mL). Primary isolated peripheral blood mononuclear cells (PBMCs) of five healthy donors were cultivated with and without additional synthetic sPecam-1 for 24h under physiological conditions. The relative LDH release of the PBMCs was determined via optical density (OD), and interpreted after subtraction of the OD of blank medium.

### 2.5. Statistics

When performing multiple pair-wise comparisons, one-way analyses of variance [19] were performed, and *p*-values were adjusted using Bonferroni’s *p*-value adjustment multiple comparison procedure. Results are presented as vertical scatter plot with mean ± standard deviation (SD), or as vertical bar graphs with mean ± standard deviation (SD). *p*-values ≤ 0.05 were considered statistically significant. Statistical analysis was performed using GraphPad prism version 5 (by GraphPad Software, Inc, San Diego, CA, USA). Receiver operating characteristic (ROC) analyses were performed using SPSS version 22 (by IBM, Armonk, NY, USA).

## 3. Results

### 3.1. Patients

A total of 58 patients were included in this study. Of these, 16 patients were scheduled for primary TKA, 20 patients underwent aseptic revision surgery, and 22 patients had to undergo surgery due to chronic or acute PJI. Of the analyzed patients, 33 were male and 25 were female, with an overall average age of 71.2 ± 8.8 years. The analyzed groups did not show significant differences in age distribution (Figure 3).

### 3.2. Quantitative Analyses

A significantly larger quantity of sPecam-1 was present under septic (chronic + acute) (mean 73.03 ng/mL, SD 22.94) compared to aseptic conditions (mean 43.95 ng/mL, SD 11.82; *p* ≤ 0.001). Furthermore, a significantly larger amount of s-Pecam-1 was observed in patients undergoing septic and aseptic revisions compared to native joints (mean 26.02 ng/mL, SD 6.48; *p* ≤ 0.001) (Figure 4). In order to further specify sPecam-1 quantities in the septic cohort, the samples were grouped into chronic (PJI ≥ 4 weeks after surgery, or with ≥3 weeks of duration of symptoms, mean 66.47 ng/mL) and acute (early postoperative infections < 4 weeks, or acute hematogenous infections with duration of the symptoms < 3 weeks, mean 78.5 ng/mL) cases of PJI. The data revealed that both subgroups contained significantly larger sPecam-1 quantities than the aseptic samples (mean 43.95 ng/mL), as well as the native samples (mean 26.02 ng/mL) (*p* ≤ 0.001). sPecam-1 quantities in chronic vs. acute did not show a significant difference.

Next, we examined whether sPecam-1 could serve as a suitable marker for the presence of PJI. Here, we performed receiver operating characteristic (ROC) analyses to define cutoff values. We combined data obtained from all septic revision surgeries (acute and chronic), and analyzed the biomarker power vs. aseptic revision cases. With a sensitivity of 82% and a specificity of 80% (area under the curve (AUC) = 0.87; *p* ≤ 0.0001), a quantitative value of 54.3 ng sPecam-1/mL synovial fluid was defined as the cutoff value for periprosthetic joint infection (Figure 4).

### 3.3. Cytotoxic and Molecular Effects of sPecam-1 on Peripheral Blood Cells

Addressing the potential role of sPecam-1 in an infectious situation, it had to be clarified if sPecam-1, in the concentrations given by the analyses from native (low), aseptic (medium), and septic (high) revision surgeries had a cytotoxic effect on peripheral cells. The data shows that sPecam-1 does not have a cytotoxic effect (PBMCs OD 0.035 ± 0.012, low sPecam-1 concentration OD 0.04 ± 0.009, medium sPecam-1 concentration OD 0.04 ± 0.007, high sPecam-1 concentration OD 0.03 ± 0.008, Figure 5).

## 4. Discussion

Defining synovial biomarkers that enable local, minimally invasive, and reliable diagnostics of PJI is an ambitious and innovative approach. Against this background, there are several studies evaluating the potential diagnostic impact of promising molecules, of which the most important seem to be α-defensin (AD), leukocyte esterase (LE), interleukin-6 (IL-6), and D-lactate (DL). With our study, we would like to draw attention to another possible marker molecule, sPecam-1, which bears a strong potential to be a reliable biomarker for PJI. The defined quantitative cutoff value of 54.3 ng sPecam-1/mL revealed a sensitivity of 82% and a specificity of 80%. Regarding the above-mentioned diagnostic options, AD, until now, represents one of the most reliable molecules for PJI detection, and was evaluated in three level 2 and two level 3 studies [20,21,22,23,24]. In their retrospective review, which included 19 PJI out of 61 arthroplasties, Bingham et al. reported a sensitivity of 100% and a specificity of 95% due to two false positive assays [20]. For total shoulder arthroplasties, Unter Ecker et al. demonstrated a sensitivity of 75% and specificity of 96% [25]. In contrast, Renz et al. found a much lower sensitivity of 54%. Due to its high specificity of > 95%, the authors concluded by using AD as a confirmatory test rather than as a screening method for PJI [16]. A clear disadvantage is that AD testing is expensive and not available in every hospital. In contrast, leukocyte esterase (LE) testing is a cheap and commonly available diagnostic tool, which is based on a colorimetric reagent pad [26]. Compared to the sPecam-1 sensitivity level of our study, which was 82%, recent studies overall reported a lower sensitivity of LE, ranging from 66% to 75% [27,28,29]. In 2018, Deirmengian et al. recommended not to use LE test strips to rule out PJI, as they often fail to detect abundant levels of LE in synovial fluid [30]. For two different LE test strips, the authors pointed out that the combined failure to detect an elevated white blood cell (WBC) count, because of either false-negative or invalid results, was 47.1% and 41.4% [30]. Nevertheless, and according to its high specificity, LE strips serve as reliable options for a secondary confirmatory rule-in test for PJI [30]. The biggest disadvantage of LE is that the reagent strip cannot be adequately read in the presence of blood or debris, which is often admixed to synovial fluid due to a joint puncture related bleeding. Against this background, different studies report an invalid result rate of 9.5–29.2% [26,27,31,32]. sPecam-1 quantification is unsusceptible to impurities, which highlights an important advantage of the present diagnostic tool.

In a study by Nilsdotter et al., IL-6 synovial analysis demonstrated a sensitivity of 69% and specificity of 93% for PJI detection of the hip [33]. A recent meta-analysis reviewed 16 studies evaluating IL-6 as a biomarker for PJI [34]. The pooled sensitivity and specificity were 83% and 91%, respectively. Compared to our results on sPecam-1, IL-6 shows a similar sensitivity, but higher specificity. However, according to the results of a study by Shin et al., an elevated proinflammatory signaling, resulting from monocytes that respond to polyethylene (PE) particles by producing IL-6 in patients with aseptic implant loosening, potentially leads to the circumstance that IL-6 has a much lower specificity of 77% compared with the findings from overall analysis (91%) [35]. The question of synovial sPecam-1 levels also corresponding to PE wear has yet to be determined.

Yermak et al. [36] evaluated the performance of synovial fluid D-lactate for the diagnosis of PJI by spectrophotometrical testing in 148 patients, of which 44 (30%) were diagnosed with PJI. By defining a cutoff at 1.26 mmol/L, a sensitivity of 86.4% and specificity of 80.8% was observed [36]. Though Yermak et al. reported a slightly higher sensitivity of D-lactate, these results are comparable to our study. Furthermore, both test methods only require a low synovial fluid volume and a short processing time.

Throughout the literature, synovial white blood cell count (WBC) and the percentage of polymorphonucleocytes (PMN%) serve as the gold standard in PJI detection. Lee et al. recently conducted a meta-analysis and reported about a pooled sensitivity of 89% and specificity of 86% for both parameters [37]. Given these results, a reliable diagnostic performance has to be stated. Nevertheless, PJI diagnosis remains challenging, especially in patients with suspected low-grade infections, as well as within the short-term postoperative course [38,39]. Although national and international guidelines consider synovial analysis of WBC and PMN% as the most important diagnostic criteria, the respective cutoff values show substantial differences [17,18,40,41]. This highlights the fact that the implementation and improvement of defined cutoff values for WBC and PMN% are the subject of current scientific debate. Given these thoughts, our study reveals a new marker molecule which shows promising results with regard to further improvements of PJI diagnostics.

This study has noteworthy limitations and leaves pending issues. We found that sPecam-1 is susceptible to storage time and temperature, but steadily and reliably regulated under infectious conditions if analyzed from fresh samples. In order to use the molecule in its biomarker function, a measurement must ideally take place immediately after the sample is taken, or within 24 h at 4 °C, or the sample needs to be stored at −80 °C for longer than 21 days in order to be able to make a valid statement. Considering the specificity (80%) and sensitivity (82%) as displayed in the ROC, when using a threshold value of 54.3 ng sPecam-1/mL synovial fluid to define an infectious status, the approach does not seem to outclass common detection methods. However, patient numbers are relatively low in this approach and need to be prospectively extended for further conclusions. According to the obtained results, it cannot be reliably determined if sPecam-1 is produced to fight pathogens, or if it can be seen as a by-product with potential unfavorable reactions. However, according to the fact that sPecam-1 does not have a cytotoxic effect on the entity of PBMCs, we do not assume that it acts as a simple byproduct.

This approach solely represents the possibility of using sPecam-1 as a biomarker for PJI. As with other biomarkers, the question arises as to whether subgroup analyses that go beyond a division into chronic and acute, e.g., considering the pathogen spectrum, can contribute to a refinement of the specificity and sensitivity values. The cutoff value can easily be adjusted in favor of sensitivity in rapid test methods in order to have a convenient and low-priced early diagnosis, e.g., for secondary confirmatory rule-in testing.

## 5. Conclusions

While a clear role of sPecam-1 in infection and inflammation could be demonstrated in this pilot study, the underlying mechanism of the molecule’s natural function in septic conditions is essential and needs to be further investigated. In this respect, the primary question that needs to be addressed is whether high local sPecam-1 concentrations are the body’s response to an infection, or if an infection can flourish due to high levels of sPecam-1. While this is not of elemental interest when defining a biomarker for a status quo, it is of crucial importance when considering a potential therapeutic approach aimed at the manipulation of molecular balances. Either way, local sPecam-1-levels serve as a promising landmark in the diagnostic workflow of PJI evaluation.

## Figures and Tables

**Figure 1 jcm-10-00612-f001:**
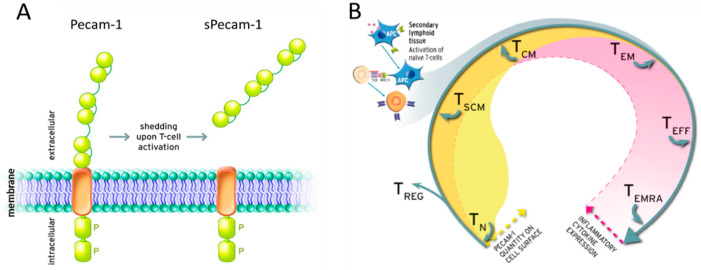
Shedding mechanism of Pecam-1 and its status in adaptive immunity. (**A**) The immunologically reactive molecule Pecam-1 is shed from the T-cell surface upon activation of immune cells. (**B**) Availability of Pecam-1 is restricted to early phases of T-cell development. Upon antigen presentation by an antigen presenting cell (APC), sPecam-1 is shed from the T-cell’s surface. The molecule’s decreasing quantity on the cell surface is illustrated by the yellow curve progression. Concomitantly but inversely, the expression of pro-inflammatory cytokines by the T-cells in various differentiation stages are rising as illustrated by the pale red curve progression. P: potential phosphorylation site, APC: antigen presenting cell, TN: naïve T-cell, TREG: regulatory T-cell, TSCN: stem central memory T-cell, TCM: central memory T-cell, TEM: effector memory T-cell, TEFF: effector T-cell, TEMRA: terminally differentiated effector memory RA T-cell.

**Figure 2 jcm-10-00612-f002:**
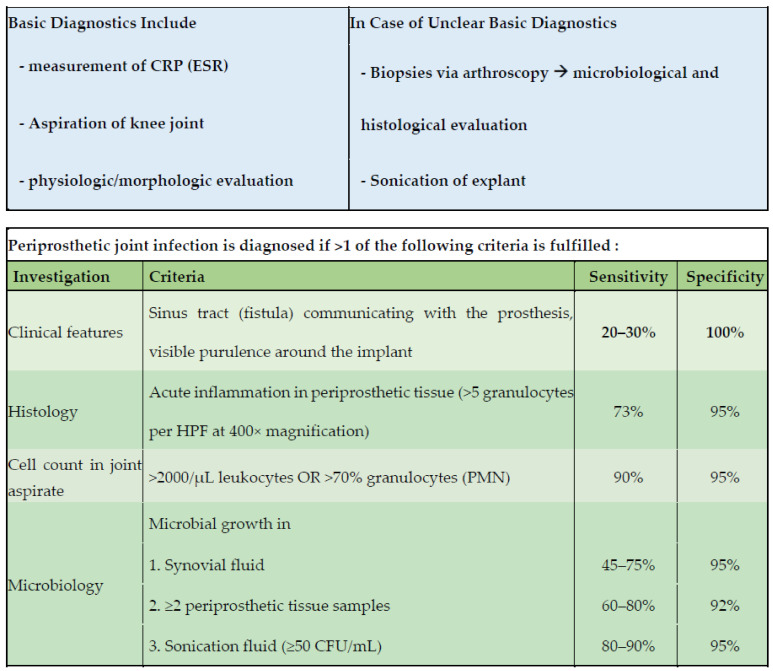
EBJIS-proposed definition of infection applied to the intervention group. Stated evaluation of sensitivity and specificity by Renz et al [16]. PJI was diagnosed when at least one of the below mentioned criteria was fulfilled. CRP: C-reactive protein; ESR: erythrocyte sedimentation rate; HPF: high-power field; PMN: polymorphonuclear granulocytes; CFU: colony-forming units.

**Figure 3 jcm-10-00612-f003:**
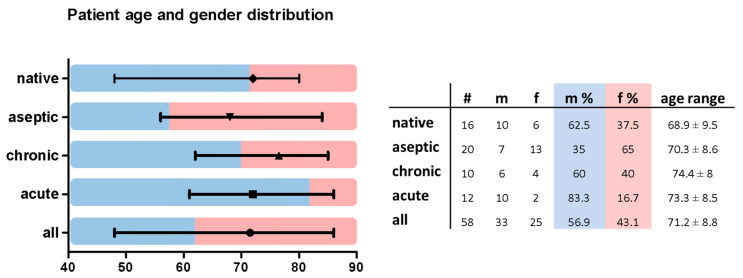
Patient group characterization by age and gender. Native, native knee joints prior to TKA; aseptic, aseptic revision TKA; chronic, septic revision surgery due to chronic PJI; acute, septic revision surgery due to acute PJI; #: number of patients; m: male; f: female.

**Figure 4 jcm-10-00612-f004:**
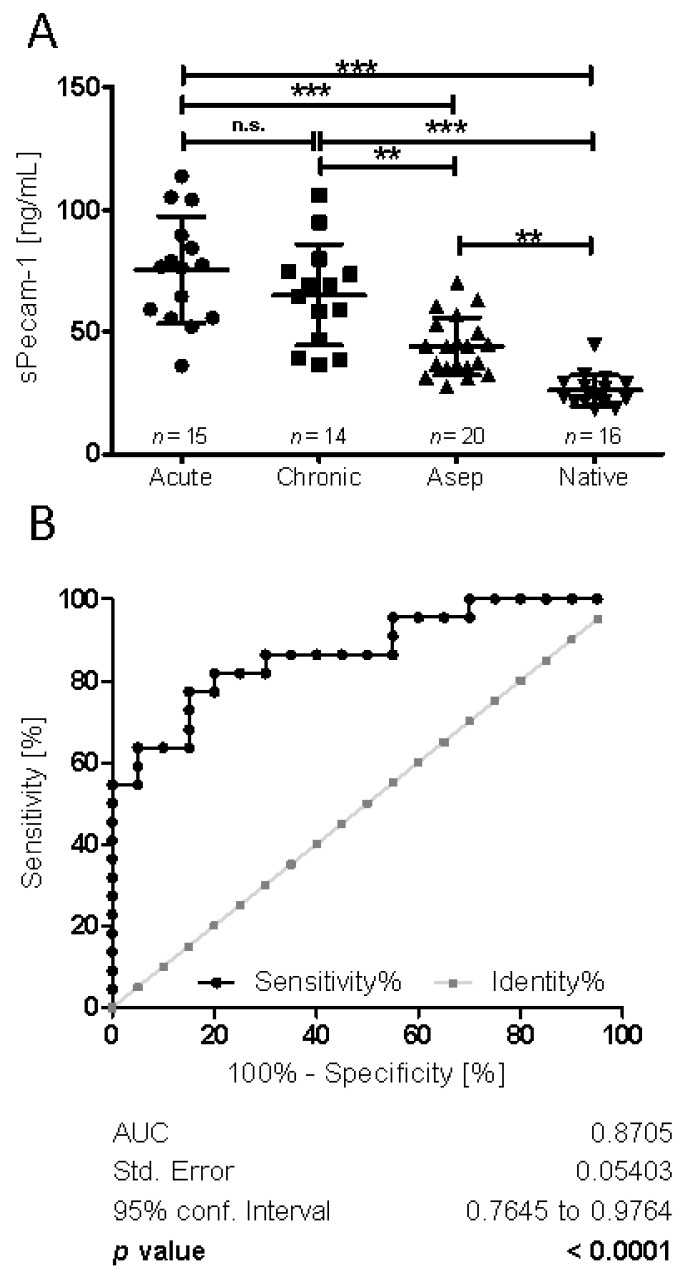
sPecam-1 quantification in samples from native, aseptic vs. septic (chronic vs. acute) revision surgeries. (**A**) Synovia samples were screened for their sPecam-1 content via Elisa, revealing significant differences between the groups according to their categorization to “septic (acute)”, “septic (chronic)”, “aseptic”, and “native”. We examined whether the marker differentiates between chronic and acute PJI. It was shown that the sPecam-1 quantities were not significantly different between chronic and acute PJI, but significantly higher for both groups compared to the aseptic, as well as to the native cohort. ***: *p* < 0.001 (very significant), **: *p* from 0.001 to 0.01 (very significant), n.s.: not significant (**B**) sPecam-1 as biomarker indicating PJI; ROC analysis with 95% confidence intervals was performed.

**Figure 5 jcm-10-00612-f005:**
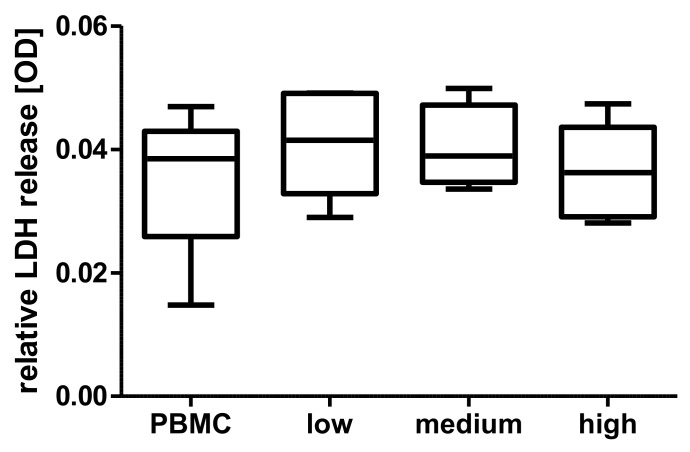
Cytotoxic molecular effects of physiological sPecam-1 concentrations. The molecule does not have any cytotoxic effect on peripheral mononuclear blood cells (PBMCs). Concentrations for cultivation were chosen according to the physiological levels measured under native, aseptic, and septic conditions (*n* = 5). LDH: lactate dehydrogenase; OD: optical density.

## Data Availability

The data that support the findings of this study are available from the corresponding author, [F.A.S.], upon reasonable request.

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
