# Peer review of "Soluble Pecam-1 as a Biomarker in Periprosthetic Joint Infection"

_jcm, 2021, doi:10.3390/jcm10040612_

Round 1
Reviewer 1 Report
A very interesting study on a new promising biomarker sPecam-1 in periprosthetic joint infections. The authors clearly presented their findings and their conclusions were correctly based on the findings of their study and the statistical analysis applied.
I recommend publishing this study in your journal.
Author Response
"Please see the attachment."

Reviewer 2 Report
Dear Authors,
thank you for submitting your paper "Soluble Pecam-1 as a Biomarker in Periprosthetic Joint Infection" to the Journal of Clinical Medicine".
Your study aims to prove the potential of sPecam-1 as a marker for PJI in prospectively collected samples in a unit for musculoskeletal and periprosthetic infections in one academic centre.
The study could show, that the local soluble Pecam-1 levels correlated to the infection status of the implant and therefore bear a strong potential to act as a biomarker of PJI. While a clear role of sPecam-1 in infection could be demonstrated, the underlying mechanism of the molecule's natural function needs to be further unraveled.
As already mentioned by you, the study has some limitations;
-First is the susceptibility of the new biomarker to storage time and temperature.
-Second, there is a relatively low number of patients included and it should be extended prospectively for further conclusions.
-Additionally, there are some errors found in the manuscript, which should be corrected before publication;
Several times in the text there is an error mentioned, because the reference source is not found. (line 84, 96, 120, 133, 149, , 159, 173, 296, ) -please clarify that. Additional there are some mistakes in the text in the discussion section, which have to be corrected. (line 199, 210, 267)
Furthermore I would recommend to add a recent paper to the introduction section, concerning the role of established inflammatory markers for diagnosing periprosthetic joint infections.
Inferior performance of established and novel serum inflammatory markers in diagnosing periprosthetic joint infections. Int Orthop. 2020 Nov 27. doi: 10.1007/s00264-020-04889-z. Online ahead of print.
Due to the originality and novelty of this study and the importance of this paper for the scientific community, I would recommend this paper for publication after considering the minor revisions mentioned above.
Best Regards
Reviewer 3 Report
The authors present a sophisticated pilot study to establish a new biomarker for periprosthetic joint infection. 58 patients prior to knee-arthroplasty were analyzed. Native knees and revision arthroplasties with either aseptic or septic history were compared. sPecam-1 was analysed by ELISA from synovial fluid samples and found significantly increased in septic conditions (n=22) compared to aseptic conditions (n=20). However, aseptic revisions also showed larger amounts of sPecam-1 compared to native knees but lower values compared to septic knees. The authors consider sPecam-1 as a new biomarker for periprosthetic infections. Unfortunately, the reasons for elevated sPecam-1 remain unclear and it cannot be stated if sPecam-1 is produced by the cells to fight infection or if it is a by-product with potential unfavorable reactions. As an advantage of the study, additional testing was performed in an in vitro cell culture study, where sPecam-1 did not reveal cytotoxic effects.
Taken together, the presented study is innovative, well performed and presented with detailed information and encouraging results. Treatment groups are well defined. Statistics are adequate and also include p-value adjustment to address multiple comparisons effects. It might be, however, problematic to combine highly different surgical approaches (DAIR, 1-stage revision, 2 stage revision) in one treatment group (“infected group” n=22). This group will be highly heterogeneous from a clinical point of view as this will have caused different surgical strategies, which will affect the results. To solve this, more clinical data and perhaps subanalyses regarding the 3 different surgical groups (DAIR, 1-stage revision, 2 stage revision) would be favorable to allow further subgroup analyses. The aseptic group would also benefit from further clinical details, as it is often difficult to distinguish infected from non-infected endoprosthesis. Did those patients with higher sPecam-1 levels in the non-infected revision group reveal clinical differences compared with those patients with lower sPecam-1 levels in this group? I would recommend to include some more clinical details from the treatment groups.
